# Crosstalk between Melanin Concentrating Hormone and Endocrine Factors: Implications for Obesity

**DOI:** 10.3390/ijms23052436

**Published:** 2022-02-23

**Authors:** Eva Prida, Sara Fernández-González, Verónica Pena-León, Raquel Pérez-Lois, Johan Fernø, Luisa María Seoane, Mar Quiñones, Omar Al Massadi

**Affiliations:** 1Instituto de Investigación Sanitaria de Santiago de Compostela, Complexo Hospitalario Universitario de Santiago (CHUS/SERGAS), Travesía da Choupana s/n, 15706 Santiago de Compostela, Spain; eva.prida@rai.usc.es (E.P.); sara.fernandez.gonzalez@rai.usc.es (S.F.-G.); veronica.penaleon@gmail.com (V.P.-L.); raquel.perez.lois@rai.usc.es (R.P.-L.); luisamaria.seoane@usc.es (L.M.S.); 2CIBER Fisiopatología de la Obesidad y Nutrición (CIBERobn), 15706 Santiago de Compostela, Spain; 3Hormone Laboratory, Department of Medical Biochemistry and Pharmacology, Haukeland University Hospital, N-5020 Bergen, Norway; johan.ferno@uib.no; 4Department of Physiology, CIMUS, University of Santiago de Compostela—Instituto de Investigación Sanitaria, 15782 Santiago de Compostela, Spain

**Keywords:** MCH, hormones, energy balance, food intake, body weight, adiposity, type 2 diabetes, obesity

## Abstract

Melanin-concentrating hormone (MCH) is a 19aa cyclic peptide exclusively expressed in the lateral hypothalamic area, which is an area of the brain involved in a large number of physiological functions and vital processes such as nutrient sensing, food intake, sleep-wake arousal, memory formation, and reproduction. However, the role of the lateral hypothalamic area in metabolic regulation stands out as the most relevant function. MCH regulates energy balance and glucose homeostasis by controlling food intake and peripheral lipid metabolism, energy expenditure, locomotor activity and brown adipose tissue thermogenesis. However, the MCH control of energy balance is a complex mechanism that involves the interaction of several neuroendocrine systems. The aim of the present work is to describe the current knowledge of the crosstalk of MCH with different endocrine factors. We also provide our view about the possible use of melanin-concentrating hormone receptor antagonists for the treatment of metabolic complications. In light of the data provided here and based on its actions and function, we believe that the MCH system emerges as an important target for the treatment of obesity and its comorbidities.

## 1. Introduction

### 1.1. MCH: Precursor, Products, and Metabolic Actions

Melanin-concentrating hormone (MCH) was first isolated in 1983 in the pituitary of chum salmon (Oncorhynchus keta), and its name comes from the ability of this hormone to control skin pigmentation [1,2,3]. Since its discovery, subsequent studies have determined that MCH is specifically expressed in the lateral hypothalamic area (LHA) and incertohypothalamic area of mammals, two important anatomic areas in the regulation of feeding behavior and energy conservation [4,5,6]. The gene encoding the pro-melanin-concentrating hormone (Pmch) generates a peptide of 165aa (pre-pro MCH) that is highly evolutionarily conserved and that, through different post-translational processes, gives rise to several products, of which MCH is the most metabolically relevant [4]. In rodents and humans, MCH comprises a 19aa peptide with a cysteine bridge [7], a molecular feature that gives MCH a cyclic structure essential for its biological activities [8]. The Pmch gene also yields other pro-MCH neuropeptides, such as neuropeptide-glutamic-isoleucine and neuropeptide-glycine-glutamic acid [4,9]. The function of these peptides remains elusive, but they seem to be devoid of metabolic effect [10,11,12]. MCH projections are widely distributed throughout the brain [4,5,6,13], suggesting that MCH could exert an array of biological functions. Indeed, MCH displays a well-characterized role in diverse physiologic processes, such as sleep [14], mood [15], and reproduction [16]. However, after the evidence presented in the mid-nineties that this peptide induces feeding [17], MCH rapidly came onto the scene as a pivotal player in the regulation of energy homeostasis. With regard to its metabolic functions, MCH induces an anabolic state/positive energy balance when centrally administered, by increasing food intake and the activation of peripheral lipid metabolism in the white adipose tissue (WAT) and in the liver [17,18,19]. In a similar manner, the over-expression of MCH in transgenic mice induces obesity by altering energy expenditure and locomotor activity [20]. Conversely, the pharmacological [21,22,23] and genetic (total or neuron-specific) inhibition of MCH [12,24,25,26,27] or melanin-concentrating hormone receptor 1 (MCH-R1) [28] leads to a negative energy balance. It is important to note that in pharmacological models, altered feeding behavior is the main cause of the metabolic actions of MCH signaling. However, in genetic mice models, the determining factor is often energy expenditure, as changes in feeding behavior are variable [2]. These divergent results could be due to regulatory changes in the genetic models caused during early development [29].

Interestingly MCH neurons also express other neurotransmitters, such as nesfatin or cocaine-and-amphetamine-regulated transcript [27,30,31]. Although the functions of these neuropeptides expressed by MCH neurons are poorly understood, they may act as mediators for the physiologic actions of these neurons [25,27,32]. In line with this, MCH neurons are mostly inhibitory and glutamatergic [13,32], although a subset was reported to be gamma aminobutyric acid (GABA)-ergic [32,33].

### 1.2. MCH Regulation

There are some preliminary reports demonstrating the presence of MCH in serum, but the source and functionality of the peripheral MCH immunoreactivity is still elusive [34,35]. However, consistent with its metabolic actions, MCH regulation changes with differences in nutritional status. Concretely, this neuropeptide is overexpressed after acute fasting [17], but not after long-term energy restriction [36]. MCH expression levels are increased in most models of obesity [37,38,39], suggesting that MCH may be involved in the etiology/pathogenesis of this disease. Interestingly, the MCH peptide content was significantly higher at the end of the sleeping period than at the end of the active period [40], suggesting that the level of the MCH peptide exhibits circadian variation. This result is in agreement with previous reports indicating that MCH levels in cerebrospinal fluid and neuronal MCH activity under normal or disturbed circadian rhythms patterns show a day/night difference [41,42,43]. However, other studies based on Pmch gene expression contradict these findings [44,45].

## 2. MCH Receptors

In 1999, MCH-R1 was characterized as the functional receptor by which MCH induces its actions. This was done by different research groups in parallel using reverse pharmacology as the main methodology [46,47,48]. The MCH-R1 is a seven-transmembrane domain G protein-coupled receptor that had previously been identified as an orphan receptor with the name of SLC-1/GPR24 [49]. In humans, the encoding gene is located in chromosome 22, q13.3. The peptide sequence comprises 353aa and is highly conserved in rats, mice, and humans (human–rat, 96% identity; human–mouse, 95% identity) [2]. MCH activates multiple intracellular signaling pathways, and these are associated with Gαi, Gαo, and Gαq/11 proteins [50]. Activation of MCH-R1 transfected in mammalian cells leads to an increase in intracellular free Ca^2+^ by an increase in phospholipase Cβ and an inhibition of forskolin-stimulated cyclic adenosine monophosphate [46]. Furthermore, MCH stimulates phosphoinositide metabolism and mitogen-activated protein kinase (MAPK) activity by both the Gαi and Gαo proteins, but in an independent manner. Therefore, protein kinase C is essential for the stimulation of MAPK when this is dependent of Gαo protein, but not when it depends on the Gαi subunit [2,50].

The MCH-R1 is primarily expressed in the brain, but is also expressed in moderate levels in peripheral tissues, such the eyes or skeletal muscles, and to a lesser extent in the tongue and pituitary tissues [46]. In the brain, the highest expression of MCH-R1 is detected in the piriform cortex and the olfactory tubercle [51,52]. Strong expression was also detected in the hippocampal formation, the shell of the nucleus accumbens (NAc), and the amygdala [51,52], suggesting a role for MCH in olfactory learning and food reward, two important elements in the regulation of feeding [3]. In line with this, an expression of MCH-R1 was found in the hypothalamus [51,52], the main homeostatic brain center that controls feeding and metabolic processes. This localization of the receptor indicates that MCH-R1 may specifically mediate the effects of MCH on appetite and metabolism.

Another MCH-R of the 340-amino-acid was identified, called melanin concentrating hormone receptor 2 (MCH-R2) [53,54]. This receptor is located in chromosome 6 at band 6q16.2-16.3 and exhibits a 38% homology with MCH-R1. Its expression is similar to that of MCH-R1, but appears to be less abundant and to be expressed in less tissues than MCH-R1 [55]. MCH-R2 displays its highest expression in the brain frontal cortex, amygdala, and NAc [53,54], but also in the peripheral tissues, including adipose, prostate, and intestine tissues [53]. The MCH-R2 only couples to the Gαq protein subunit [54]. Interestingly this MCH-R2 is expressed only in certain mammals such as ferrets, dogs, or humans [56,57], but is not expressed in rodents [57], and its regulation in a gain-of-function mouse model showed opposite metabolic actions when compared to the MCH-R1 [13].

## 3. Crosstalk of MCH and Endocrine Factors

It is well known that the neuronal network in the hypothalamus may integrate peripheral information, such as nutrients and hormones, to modulate energy balance [58,59,60]. In line with this, some of the classical endocrine factors such as leptin, ghrelin, growth hormone (GH), luteinizing hormone (LH), cortisol or thyroid hormone (TH) that are well-known regulators of energy homeostasis, interact closely with MCH to exert different physiological functions. It is important to note that impairment of these signaling cascades could lead to obesity and related comorbidities, such as type 2 diabetes, non-alcoholic fatty liver disease, or some types of cancer [61,62]. Due to the rise in the prevalence of obesity, it has become a major social, economic, and health problem [63,64,65]; therefore, it is important to preserve the proper functionality of the endocrine physiology to avoid any metabolic complications.

### 3.1. MCH and Insulin

There is a variety experimental evidence showing an interaction between MCH and insulin. For example, a peripheral administration of insulin produces an increase in the expression of mRNA and protein levels of MCH [66], as well as an increase in the excitability of the MCH neurons [67] (Figure 1: pp 6). Likewise, it has been shown that insulin signaling in MCH neurons is dispensable for energy and glucose homeostasis in mice fed a standard diet. However, mice deficient for the insulin receptor (IR), specifically in the MCH neurons (IRDMCH), have a greater sensitivity to insulin compared to control mice when fed a high-fat diet (HFD) [67]. These IRDMCH mice had lower glucose production in the liver, while the uptake of glucose in WAT or brown adipose tissue (BAT) is not affected, indicating that the alteration of insulin signaling in the MCH neurons of obese mice improves their ability to suppress the production of hepatic glucose, producing a greater sensitivity to insulin [67].

### 3.2. MCH and Leptin

Like insulin, leptin also modulates the activity of the MCH system [68] (Figure 1). MCH expression increases in obese leptin-null (ob/ob) mice, suggesting a close interplay between MCH and leptin activity. In line with this, obese rats with a point mutation in the long form of the leptin receptor, the Zucker LepR fa/fa rats, exhibit an upregulation of MCH expression and a downregulation of MCH-R1 levels in comparison with lean rats [38]. At the same time, the fasting-dependent decrease in serum leptin induces an increase in the circulating MCH levels that, in turn, are counter-regulated by the administration of human recombinant leptin [69]. If we take these data together, we can deduce that the lack of leptin is associated with an increase in the hypothalamic expression of MCH, which is consistent considering that MCH stimulates food intake and promotes an increase in body weight [3]. Nevertheless, MCH transgenic mice are hyperleptinemic [20], and neural MCH knockout (KO) mice display hypoleptinemia. However, the leptin response on appetite in these mice is normal [25].

In agreement with these data, when deleting MCH-R1 in ob/ob mice, a phenotype characterized by greater lean mass, lower body fat and greater locomotor activity is achieved compared to ob/ob mice that present intact MCH-R1 [70]. However, the congenital disruption of the MCH peptide or the loss of MCH neurons in ob/ob mice leads to an improvement of the obese phenotype [25,71]. Of note, this did not occur when the disruption of the MCH system was performed in adulthood [72], where the loss of MCH only induces an improvement in glucose homeostasis. Therefore, these double congenital KO mice are hypophagic, lean, and show low glucose levels compared to ob/ob mice. However, the reversal of obesity is not complete, indicating that other pathways are involved in the response to chronic leptin deficiency. The greater amount of lean mass and the lower level of fat in the MCH-R1-deficient ob/ob mice could be explained as a consequence of the increased locomotor activity, which in turn would explain the greater sensitivity to insulin. All of these results suggest an important role of MCH in mediating the physiological response of chronic leptin deficiency.

### 3.3. MCH and Somatotropic Function

MCH and the gastric hormone ghrelin take part in the regulation of energy homeostasis, and both stimulate food intake [17,58]. Ghrelin administration or ghrelin depletion by a surgical intervention does not regulate the MCH mRNA expression in the hypothalamus [73]. Consistent with this, the orexigenic action of ghrelin is independent of MCH signaling since ghrelin is able to induce food intake in MCH-R KO mice [74]. However, MCH signaling seems to play a role for other ghrelin actions, such as the induction of GH secretion [74,75]. Thus, GH mRNA levels were markedly increased in response to ghrelin injection in the wild-type (WT) mice, but were blunted in the MCH-R KO mice [74]. In line with this, it has been reported that MCH-R1 is expressed in the pituitary gland of rodents and humans [46,76], and that MCH neurons project to the median eminence region of the brain [5,77]. These data indicate that the MCH system affects pituitary function. The functional characterization of this interaction was determined by studies showing that MCH increased GH-secretion in human and rodent pituitary cells [75]. Furthermore, chronic in vivo MCH administration results in an increase in plasma insulin growth factor 1 (IGF-1) levels [78]. Meanwhile, MCH-R KO mice had significantly lower serum IGF-1 levels than the WT mice [74]. Finally, since IGF-1 is released mainly by the liver and, together with GH, induces changes in body weight and body composition, we can conclude that MCH may exert some of its effects on energy balance, at least in part, via regulation of the GH-IGF axis or somatotropic function (Figure 1).

### 3.4. MCH and Hypothalamic-Pituitary Gonadal Axis

Food restriction upregulates the MCH expression [17], and in turn, this hormone decreases energy expenditure and increases food intake [3]. In concordance, as we pointed out earlier, pharmacological or genetic MCH inhibition in mice has opposite effects, displaying a negative energy balance. Starvation has negative consequences on fertility because decreased circulating gonadotropins induces anovulation [79]. However, despite these effects, MCH [24] and MCH-R1 null mice [28] remain fertile. These data suggest that MCH could play a role in reproduction. In line with this, MCH fibers are in close apposition with the hypothalamic gonadotropin releasing hormone (GnRH) neurons, and 50–55% of rat GnRH neurons express MCH-R1 [80]. Thus, MCH may target GnRH neurons in the hypothalamus. Interestingly, MCH stimulates the release of luteinizing hormone-releasing hormone [Figure 1] and gonadotropins in vitro, acting at both median eminence (ME) and pituitary levels [81]. Moreover, when administered into the ME and medial pre-optic area (MPOA), MCH stimulates the release of luteinizing hormone (LH) in vivo as well [82,83]. In contrast, the administration of anti-MCH antiserum in the MPOA inhibits LH release [82]. Another approach using estradiol benzoate on ovariectomized rats confirms these results and indeed demonstrates that the effects of MCH on fertility are mediated by the melanocortin system. In this model, the effect of MCH on LH was blocked by the administration of the melanocortin 4/5 (MC4/5) antagonist and the melanocortin 3/5 (MC3/5) antagonist, but not by the melanocortin 3/4 (MC3/4) antagonist, into the MPOA [83].

In contrast, leptin, an anorexigenic hormone secreted by adipose tissue, also influences sexual maturation and fertility [84]. Both MCH and leptin stimulate sexual behavior, and it has been shown that the effect of leptin on LH release is mediated by MCH [83]. Thus, an administration of an MCH antiserum into the MPOA prevented the surge in LH induced by leptin when this hormone was injected into the zona incerta [83]. In addition, MC3/5 and MC4/5, previously shown to inhibit the stimulatory effect of MCH on LH release, also inhibited the effect of leptin [83]. These data suggest that MCH *per se* plays a relevant role in fertility, and that its activity is necessary for other hormones, such as leptin, to promote reproductive activity.

#### Role of Estrogens

Estrogens constitute a group of sex hormones derived from cholesterol (steroids) that are mainly produced in the ovaries, adrenal glands, and the placenta in pregnant women. It should be noted that estrogens are also present in low concentrations in men. Their synthesis is stimulated by the release of follicle-stimulating hormone (FSH), a hormone that regulates GnRH and that is subject to regulation by estrogens via a negative feedback loop. Estrogen levels in females vary cyclically and generally reduce food intake [85].

It has been shown that the orexigenic effect of MCH is sexually dimorphic and that estrogen is the potent responsible anorexigenic signal [86,87,88]. This affirmation is based on experiments in rats submitted to ovariectomy (OVX) and estrogen replacement. Thus, OVX rats with estrogen replacement are less sensitive to the orexigenic action of MCH compared to OVX controls and male rats [16,87,88]. Moreover, the food intake induced by MCH is clearly compromised in female rats on the estrus cycle compared to the control rats on the diestrus cycle [16]. These studies demonstrate the deleterious effect of estrogen on the MCH control of feeding and hence could explain one of the mechanisms by which the loss of estrogen in aged women induces an increase in body weight and adiposity.

### 3.5. MCH Hypothalamo-Pituitary Thyroid Axis

The thyroid axis is an important regulator of energy homeostasis, specifically with regard to the control of food intake and energy expenditure [89,90]. Fasting leads to a profound suppression of the thyroid stimulating hormone (TSH) in the pituitary, thyroid releasing hormone (TRH) in the paraventricular nucleus (PVN), as well as circulating thyroxine [91]. Due to the well-known effects of calorie restriction on MCH and the fact that MCH-R1 is expressed in thyroid follicular cells, an interaction between these endocrine systems is plausible. In this sense, mice lacking MCH-R1 exhibit reduced secretion of the circulating inactive form of thyroid hormone (TH), namely the free and total tetraiodothyronine and the active form of TH or liothyronine. At the same time, these mice display high levels of TRH and TSH when compared with WT mice [28,92]. Moreover, when MCH is administered intracerebroventricularly, plasma TSH in vivo and the release of TRH from hypothalamic explants are significantly reduced [93] (Figure 1). Furthermore, MCH was also shown to significantly reduce the TRH-stimulated TSH release from pituitary cell cultures [93]. These data suggest a possible role of MCH in the control of energy homeostasis via the inhibition of the thyroid axis. Specifically, these data could explain the mechanism responsible for the decreased energy expenditure after MCH injection.

### 3.6. MCH and Hypothalamic-Pituitary Adrenal Axis

The hypothalamic pituitary adrenal (HPA) axis is a neuroendocrine pathway that is activated by stress and comprised of three endocrine components: (1) the corticotrophin-releasing factor (CRH) that resides within the PVN, (2) adrenocorticotropic hormone (ACTH) that is released from the anterior pituitary, and (3) cortisol and corticosterone which are the main glucocorticoids secreted by human and rodent adrenal glands, respectively [94]. The direct administration of MCH to hypothalamic explants stimulates the release of CRH (Figure 1), an effect that is blocked by the administration of the MCH-R1 antagonist. In vivo, a specific injection of MCH into the PVN increases circulating serum levels of ACTH and corticosterone [95,96,97]. However, MCH reduces corticosterone levels in rats after exposure to stress [97]. Moreover, MCH has an anxiogenic-like effects when it is administered centrally, in agreement with the elevated MCH levels observed after being subjected to the elevated plus maze or the forced swim test, two standard test to measure the anxiety/stress response in rodents [97,98]. In contrast, MCH-R1 antagonist exhibits opposite effects and attenuates the anxiogenic and prodepressant activity in rodents [97,98]. This interesting effect of MCH on stress and depression provides an advantage for the development of a therapeutic option targeting MCH signaling.

## 4. Conclusions and Remarks

The prevalence of obesity is rising, reaching pandemic proportions in both developed and developing countries. Obesity may lead to comorbidities such as type 2 diabetes, cardiovascular disease, and liver disease, making it is a complex and multifactorial illness involving genetic, biological and behavioral factors. Substantial progress has been made in the last few decades in understanding the molecular pathways and physiological systems governing energy balance. While several anti-obesity drugs have been approved by the Food and Drug Administration and/or the European Medicines Agency, their efficacy and/or safety is still a concern. Currently, the most effective and sustainable approach to decrease body weight and to improve glucose metabolism is bariatric surgery; however, its invasiveness and side effects limits the access of this approach to obese patients in the general population. Because of this, the elucidation of new therapies that could help to slow, prevent, or revert the obesity epidemic are urgently needed. In this sense MCH, due to its actions on food intake, adiposity, energy expenditure, and food motivation, has attracted the attention of several pharmaceutical companies as a possible target for the development of anti-obesity drugs. As we pointed out previously, MCH, in addition to increased feeding, regulates glucose homeostasis and insulin sensitivity, suggesting its potential as a target for the combined treatment for both obesity and type 2 diabetes. Furthermore, MCH expression levels are increased in several obese models, such as agouti mice, leptin and leptin receptor deficient mice, and Zucker rats, suggesting that MCH may be involved in the etiology/pathogenesis of this disease. Importantly, an MCH-R1 antagonist could be effective in pathological states associated with age, such as estrogen deficiency that occurs in women after menopause, or in improving leptin sensitivity in states associated with leptin resistance.

It is important to note that the development of compounds based on MCH-R1 ligands present some advantages compared to other potential drugs. Using MCH-R1 antagonists in preclinical animal models has shown efficacy against anxiety and depression due to the effects on the adrenal axis. This is contrary to other compounds, such as rimonabant or lorcaserin that were withdrawn from the market for their off-target consequences leading to psychological side effects. Despite the promising data obtained from preclinical animal models, only six MCH-R1 antagonists underwent clinical trials. Unfortunately, all of them were stopped in phase I clinical trials for several reasons, including low safety, low bioavailability, or lack of efficacy. Therefore, these features should be improved with the next generation of compounds, as we believe that MCH-R1 is still a promising target for the treatment of metabolic diseases. In this sense, other drugs based on MCH signaling with the intent to treat metabolic derangements other than obesity should be explored. Candidates for treatment could be anorexia or liver disease, for which this peptide was found to have therapeutic potential. Furthermore, due to the high orexigenic and adipogenic action of MCH, the therapeutic use of MCH agonist should be considered in the treatment of other diseases characterized by the chronic loss of lean or fat mass, such as sarcopenia or cachexia.

Importantly, increasing the knowledge of the interaction of MCH with other hormones can help to design more effective antagonists, as well as avoid adverse side effects. Besides this, it could be relevant to combine contemporary therapies intended to target various targets at once in combinatory therapy or unimolecular polypharmacy. Moreover, due to the neuron-specific action of MCH on metabolic processes such as peripheral lipid deposition, induction of glucose intolerance, or feeding, the use of novel technologies that target specific neuron populations in the brain could provide new avenues in the development of MCH based therapies for treating metabolic diseases. In this sense, the direct inhibition of MCH at the neuronal level instead of the broad pharmacological inhibition of MCH-R1 seems a more plausible and effective therapeutic method due to its specificity. However, this strategy is still elusive and will require development of new targeted molecules or delivery methods.

Overall, after studying the relationship of MCH to other endocrine factors and how they influence metabolic control as a whole, we believe that the study of this hormone, as well as pharmacological exploration of its potential as a drug, is of utmost importance to allow us to tackle obesity and its related complications.

## Figures and Tables

**Figure 1 ijms-23-02436-f001:**
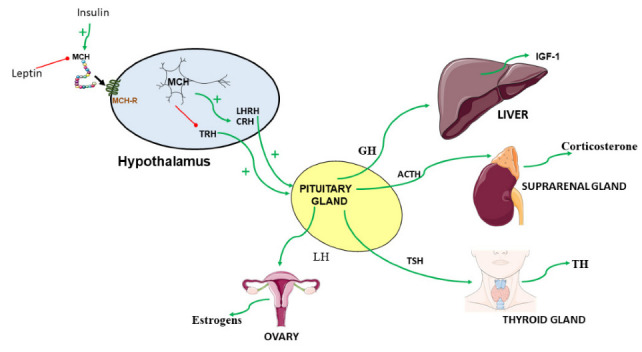
A description of the crosstalk between the MCH and endocrine factors. Abbreviations used: ACTH—adrenocorticotropic hormone; CRH—corticotrophin-releasing factor; GH—growth hormone; IGF-1—insulin growth factor 1; LH—luteinizing hormone; LHRH—luteinizing hormone releasing hormone; MCH—melanin concentrating hormone; MCH-R—melanin-concentrating hormone receptor; TH—Thyroid hormone; TRH—thyroid releasing hormone; TSH—thyroid stimulating hormone. The figures were generated by using materials from Servier Medical Art (Servier) under consideration of a Creative Commons Attribution 3.0 Unported License.

## Data Availability

Not applicable.

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
