# Peer review of "Crosstalk between Melanin Concentrating Hormone and Endocrine Factors: Implications for Obesity"

_ijms, 2022, doi:10.3390/ijms23052436_

Round 1
Reviewer 1 Report
In the manuscript “Crosstalk Between Melanin Concentrating Hormone and Endocrine Factors: Implications for Obesity”, the authors reviewed and summarized the current knowledge of the crosstalk of melanin concentrating hormone with different endocrine factors, as well as its implications for obesity. The paper is interesting and relevant. Nevertheless, I have some suggestions that could improve the scientific outcome of the manuscript.
Minor comments:
In my opinion, the paper needs some English spelling check.
The title mentions obesity. However, the first direct mention to obesity occurs only in the conclusion. It would be interesting to add a paragraph about obesity and other relevant metabolic disorders where appropriate, namely when mentioning hormones whose levels are impaired in these conditions.
Abstract:
- The authors should state the full name before introducing abbreviations (eg. MCH, LHA and BAT)
- Instead of or after “large number of physiological functions”, the authors could state 2 or 3 examples
Estrogens: It would be important to mention that estrogens are also present in men. This may be misunderstood as the text mainly mentions ovaries ans effects on women.
Conclusion
The conclusion lacks references about obesity prevalence and its associated pathologies. Moreover, obesity not always leads to the development of these comorbidities. It is safer to say that it “may lead to” as sometimes diabetes appears first than obesity for instance.
Author Response
Referee comments to author:
Referee 1:
In the manuscript “Crosstalk Between Melanin Concentrating Hormone and Endocrine Factors:
Implications for Obesity”, the authors reviewed and summarized the current knowledge of the crosstalk of melanin concentrating hormone with different endocrine factors, as well as its implications for obesity. The paper is interesting and relevant. Nevertheless, I have some suggestions that could improve the scientific outcome of the manuscript.
RESPONSE: We would like to thank the Reviewer for the positive and encouraging comments on our manuscript.
Minor comments:
In my opinion, the paper needs some English spelling check.
RESPONSE: We apologize for these mistakes. We have sent the manuscript to a professional native English speaking and the grammar and spelling of the manuscript was thoroughly corrected
The title mentions obesity. However, the first direct mention to obesity occurs only in the conclusion. It would be interesting to add a paragraph about obesity and other relevant metabolic disorders where appropriate, namely when mentioning hormones whose levels are impaired in these conditions.
RESPONSE: We thank the Reviewer for this important comment. To clarify this aspect, we have now added a paragraph in section 3 (new line 130).
Abstract:
• The authors should state the full name before introducing abbreviations (eg. MCH, LHA and BAT)
• Instead of or after “large number of physiological functions”, the authors could state 2 or 3 examples
RESPONSE: We thank the Reviewer for these comments, and has following this
recommendation. Due to the referee 2 had raised the same concern the same comment we have decided following his/her recommendation to remove the abbreviations from the abstract.
Estrogens: It would be important to mention that estrogens are also present in men. This may be misunderstood as the text mainly mentions ovaries ans effects on women.
RESPONSE: We agree with this comment, and following the reviewer’s recommendation, we have included modifications in the text.
Conclusion
The conclusion lacks references about obesity prevalence and its associated pathologies. Moreover, obesity not always leads to the development of these comorbidities. It is safer to say that it “may lead to” as sometimes diabetes appears first than obesity for instance.
RESPONSE: We thank the Reviewer for these interesting comments. However, because referee 3 proposes to remove the references from the conclusions section, we have added these references in the new line 132 and made some changes in the text.

Reviewer 2 Report
This manuscript titled "Crosstalk Between Melanin Concentrating Hormone and Endocrine Factors: Implications for Obesity" by Prida E. et al. nicely reviewed the state-of-the art literature on MCH signaling, its interaction with different endocrine factors and impact of MCH signaling on obesity. It was a pleasure reading this manuscript which is well-organized, concise and reviewed an important topic.
I only have some minor comments which is mostly about English, grammar and spelling.
Below are specific points:
- Please define or avoid all the acronyms in the abstract. This journal has a broad readership and undefined or non-standard acronyms in the abstract makes it harder for the readers to get the message.
- A general point about the pharmacological inhibition of MCH or MCH-R1: which one would be a more viable approach to achieve therapeutic target against obesity and why ? Do the authors have any opinion on this?
- line 55, "It is important.. please correct the sentence.
- Line 72, there is an extra "s". should be removed.
- Line 151-153: The author said "we performed..." Not sure is the paper (ref. 65) was authored by the author of this manuscript. If not, this sentence should be revised.
- Line 216 and 219-220- using E as an acronym for Estrogen is not standard in the field and readers might confuse with vitamin E. I suggest using full word "Estrogen" and remove the "E" acronyms throughout the paper.
- Line 254and line 259- "MCR-R1" should it be MCH-R1?
- Fig.1 spelling error in "suprarenal gland"
- Line 293- which obese models? please specify
- Line 301- please replace "withdrawal" by withdrawn"
- Line 303- Grammar: not "undergo" should be "underwent"
- Line 304- not "by several reasons" please write "for several reasons"
Author Response
This manuscript titled "Crosstalk Between Melanin Concentrating Hormone and Endocrine Factors: Implications for Obesity" by Prida E. et al. nicely reviewed the state-of-the art literature on MCH signaling, its interaction with different endocrine factors and impact of MCH signaling on obesity. It was a pleasure reading this manuscript which is well-organized, concise and reviewed an important topic.
RESPONSE: We would like to thank the Reviewer 2 for the thoughtful comments and the
positive view of our manuscript.
I only have some minor comments which is mostly about English, grammar and spelling.
Below are specific points:
- Please define or avoid all the acronyms in the abstract. This journal has a broad readership and undefined or non-standard acronyms in the abstract makes it harder for the readers to get the message.
- A general point about the pharmacological inhibition of MCH or MCH-R1: which one would be a more viable approach to achieve therapeutic target against obesity and why ? Do the authors have any opinion on this?
- line 55, "It is important.. please correct the sentence.
- Line 72, there is an extra "s". should be removed.
- Line 151-153: The author said "we performed..." Not sure is the paper (ref. 65) was authored by the author of this manuscript. If not, this sentence should be revised.
- Line 216 and 219-220- using E as an acronym for Estrogen is not standard in the field and readers might confuse with vitamin E. I suggest using full word "Estrogen" and remove the "E" acronyms throughout the paper.
- Line 254and line 259- "MCR-R1" should it be MCH-R1?
- Fig.1 spelling error in "suprarenal gland"
- Line 293- which obese models? please specify
- Line 301- please replace "withdrawal" by withdrawn"
- Line 303- Grammar: not "undergo" should be "underwent"
- Line 304- not "by several reasons" please write "for several reasons"
RESPONSE: We apologize for these mistakes which are corrected in the revised manuscript.
We have sent the manuscript to a professional native English speaking and the grammar and spelling of the manuscript was thoroughly corrected.
Since all the reviewers have the same concern, we have decided to delete the abbreviations from the abstract.
Following the reviewer’s advice, we have also included a short paragraph where we express our opinion on the viability of the different approaches intended to target MCH system (new line 340).

Reviewer 3 Report
Narratives
- Abstract section: Expand MCA, LHA and BAT for the first time
- Conclusions needs to be revised in the last paragraph of the article.
- Conclusion section should not contain references.
Author Response
Referee comments to author:
Reviewer 3:
RESPONSE: We thank reviewer for the time dedicated to review this manuscript and their
comments
Narratives
1. Abstract section: Expand MCA, LHA and BAT for the first time
RESPONSE: We apologize for this mistake, which has been amended. In fact following the
recommendation of the referee 2 we have decided delete the abbreviations in the abstract.
2. Conclusions needs to be revised in the last paragraph of the article.
RESPONSE: This paragraph has been re-phrased and we hope it is now clearer.
3. Conclusion section should not contain references.
RESPONSE: We have changed the references and text accordingly

Round 2
Reviewer 1 Report
After the first round of revisions, I think that the paper is now ready to be published
Reviewer 2 Report
I agree with the changes and responses provided by the authors.
Reviewer 3 Report
None